# Large-Scale Screening of Thiol and Fermentative Aroma Production during Wine Alcoholic Fermentation: Exploring the Effects of Assimilable Nitrogen and Peptides

**Camille Duc [1], Faïza Maçna [1], Isabelle Sanchez [2], Virginie Galeote [1], Stéphane Delpech [3], Anthony Silvano [4] and Jean-Roch Mouret [1,*]**

[1] SPO, INRAE, University of Montpellier, Montpellier SupAgro, 34000 Montpellier, France; camille.duc@inrae.fr (C.D.); faiza.macna@inrae.fr (F.M.); virginie.galeote@inrae.fr (V.G.)

[2] MISTEA, INRAE, University of Montpellier, Montpellier SupAgro, 34000 Montpellier, France; isabelle.sanchez@inrae.fr

[3] Nyseos, 53 Rue Claude François, 34080 Montpellier, France; stephane@nyseos.fr

[4] Lallemand SAS, 19 Rue des Briquetiers, 31702 Blagnac, France; asilvano@lallemand.com

[*] Correspondence: jean-roch.mouret@inrae.fr; Tel.: +33-4-9961-2274

**Abstract:** In alcoholic fermentation, under oenological conditions, the environmental parameters impacting fermentation kinetics and aroma production have been widely studied. The nitrogen content of grape must was found to be one of the most important parameters for both of these aspects of fermentation. Many studies have been performed on the effect of mineral nitrogen addition. However, it has increasingly been observed that the nature of the nitrogen added leads to different results. Our work focused on the effects of peptide addition on both fermentation kinetics and aroma production. Peptides are one of the less well understood sources of assimilable nitrogen, as their incorporation by yeast remains unclear. In this study, we compared the effect of the addition of a "classic" assimilable nitrogen source (ammonium + amino acids) with that of peptide addition in both white and red must fermentation by screening 18 *Saccharomyces cerevisiae* strains in total. Our data show that peptide addition enhances fermentation kinetics and leads to specific changes in the production of fermentative aromas. The impact of peptides on thiol synthesis is rather limited.

**Keywords:** nitrogen; peptide; *Saccharomyces cerevisiae*; alcoholic fermentation; aromas; thiols

## 1. Introduction

Grape must is a very complex medium. It is composed of numerous nutrients that are essential for the alcoholic fermentation process, such as vitamins, lipids and nitrogen [1–3]. Nitrogen is one of the most important of these nutrients and has thus been widely studied in recent decades, and its influence on the rate or duration of fermentation has been described. Nitrogen is present in different forms in must, including proteins, peptides, free amino acids and ammonium [4]. Free amino acids and ammonium are the two nitrogen sources considered to be assimilable by yeast and are therefore the best described. However, small peptides (<1 kD) can also be found in grape must (accounting for up to 17% of the total nitrogen) and could represent up to 10% of the assimilable nitrogen present [5,6]. The effects of these specific compounds on alcoholic fermentation are still poorly documented, although they can be assimilated as different small peptide transporters are present in the yeast *S. cerevisiae*. Indeed, two di/tripeptide transporters have been identified, encoded by the *PTR2* and *DAL5* genes [7,8], as well as two tetra/penta-peptide transporters, encoded by *OPT1* (which is also a glutathione transporter) and

*OPT2*. In addition, a large number of wine strains have acquired, via an horizontal gene transfer, the *FOT1-2* genes, which have been found to be involved in the uptake of peptides up to nine amino acids, substantially expanding the number and nature of the transported oligopeptides [6,9,10].

Moreover, nitrogen metabolism is involved in many fermentative aroma production pathways. Indeed, fermentative aromas such as higher alcohols and acetate esters are produced through a well-known metabolic pathway: the Ehrlich pathway [11]. The production of these aromas is thus a function of nitrogen availability, among several other environmental factors, such as temperature and lipid content [1,12,13]. The addition of "classic" assimilable nitrogen sources (free amino acids and ammonium) during oenological fermentation has been found to increase the production of fermentative aromas [13–16]. Depending on the nature of the nitrogen added (amino acids versus ammonium), the impacts on the volatile compounds produced during fermentation are different [13,17].

Another family of aromas that may be impacted by nitrogen availability is thiols, which are varietal aromas [18]. These compounds are not produced through yeast metabolism but from odorless precursors contained in must that are released by yeast. These compounds are present in low concentrations in wine, but their perception thresholds are very low [19]. Therefore, thiols play an important role in wine aromas, especially in Sauvignon Blanc wine, in which they are well described [20]. They can also be found in red wine, but in smaller concentrations [21,22]. Most of the research concerning thiols has focused on the development of quantification methods and the identification of new precursors, but it has been shown that thiol release can be modulated by nitrogen catabolite repression [23].

The aim of the present work is to evaluate and differentiate the effect of small peptides and classic assimilable nitrogen sources on the fermentation kinetics and production of aromas. To do so, we used a novel robotized fermentation platform to set up a study aimed at characterizing a large pool of yeast strains for the fermentation of white (Sauvignon Blanc) and red (Merlot) grape musts supplemented with either small peptides or assimilable nitrogen (amino acids and ammonium). Understanding the impact of these nitrogen compounds on the metabolism of *S. cerevisiae* in alcoholic fermentation will allow better management of these nutrients to optimize volatile compound production during alcoholic fermentation. Finally, it is important to note that, despite being naturally present in grape musts and organic products derived from yeasts, the addition of small peptides alone is a prohibited practice in oenology. Therefore, our work constitutes a proof of concept of the impact of small peptides in alcoholic fermentation. The obtained results will not be directly applicable, but they will be helpful to design a strategy of organic nitrogen supply via the addition of products derived from yeasts.

## 2. Materials and Methods

### 2.1. Yeast Strains

All the yeast strains used in this study are commercial *Saccharomyces cerevisiae* strains from Lallemand SA, Montreal, QC, Canada. 10 yeast strains were used for white wine fermentation (called B1 to B10), and 8 other yeast strains were used for red wine fermentation (called R1 to R8).

Fermentation tanks were inoculated with 20 g/hL of active dry yeast that was previously rehydrated for 30 min at 37 °C in 50 g/L of glucose solution (1 g of dry yeast diluted in 10 mL of this solution).

### 2.2. Fermentation Media

Two grape musts collected in 2017 were used in this study to carry out the fermentation: a Sauvignon Blanc must from Val de Loire, France, and a Merlot must from Aude, France. The Sauvignon Blanc must contained 200 g/L sugar and 75 mg N/L assimilable nitrogen. The Merlot must contained 236 g/L sugar and 87 mg N/L assimilable nitrogen. These two grape musts were first adjusted to 150 mg/L assimilable nitrogen using a mixture of amino acids and ammonium. This nitrogen source mixture is composed of ammonium salt and amino acids in a proportion similar to that in real musts [24]. For both the white and red musts, the obtained media were considered the control must. Then, two

nitrogen conditions were set up. In the YAN (Yeast Assimilable Nitrogen) condition, 50 mg N/L of yeast-assimilable nitrogen was added using the same nitrogen mixture employed previously, whereas in the Peptide condition, 50 mg N/L of assimilable nitrogen was added, 70% of which came from the mixture of ammonium salt and amino acids, while 30% was in the form of peptides (obtained via a process specified in the next section).

### 2.3. Small Peptide Preparation

For technical reasons (i.e., the very low concentrations in the grape must, the difficulty of extraction and purification), the peptides used in this study did not come from grape must but were a result of bovine serum albumin (BSA) digestion.

BSA (46 g/L) at pH 8.5 was digested using 61.23 mg of trypsin at 37 °C for 11 h. The digestate was then purified, retaining only the fraction <1 kDa (peptides from 2 to 9 amino acids) using a 1 kDa column (5GEH-Calthcare, hollow-fiber cartridge filter, VFP-1-C-4M). Then, the purified solution was freeze dried. Measurements via the Kjeldahl method [25] indicated that the concentration of assimilated nitrogen in the obtained powder was 0.112 mg/mg of peptide powder.

### 2.4. Fermentation Conditions

Fermentation was carried out in 300 mL fermenters at 18 °C for the Sauvignon Blanc must and 26 °C for the Merlot must.

Fermenters were placed in support guides on magnetic stirring (260 rpm) plates (Figure 1). Fermentation was followed via the automatic monitoring of the released $CO_2$ every hour. This task was performed with a robotic arm (LabServices) capable of moving the fermenters successively from their location on one of the stirring plates to a precision balance to measure the weight every hour. Internally developed control software enables the users to define the experimental settings. This information is transmitted to the control application that controls the robotic arm and records the weights. The software then calculates, for each time point, 1/ the amount of $CO_2$ that is released (expressed in g/L), which is proportional to the amount of sugars that has been consumed at that time, and 2/ the fermentation rate, which corresponds to the rate of $CO_2$ production, in g $CO_2$/L.h (proportional to the rate of sugar consumption). Data are stored in a relational database and visualized using a devoted graphical interface [26].

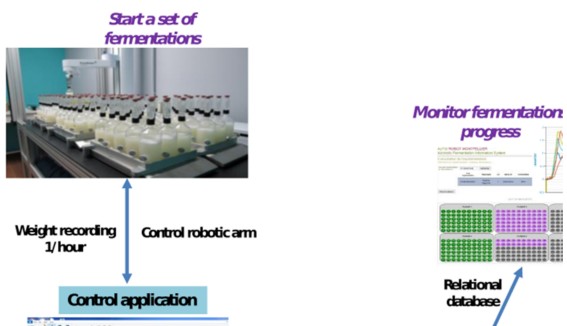

**Figure 1.** Automated robotic system for monitoring high-throughput fermentation.

To avoid the release of compounds due to cell death and/or the loss of volatile compounds, we chose to stop fermentation when 80 g/L $CO_2$ was produced.

As experiments were quite cumbersome to perform, each fermentation was performed in biological duplicate. However, in previous studies, it was shown that quadruplicate experiments run with this online monitoring system yielded highly reproducible results [12,27]. The relative standard deviation was equal to 1% for the parameters of fermentation kinetics, [27], 6% for the metabolites of the central carbon [12,27], 11% for the fermentative aromas [12] and 7% for the volatile thiols [27].

## 2.5. Determination of the Parameters of Fermentation Kinetics

From the online measurements of $CO_2$ production, several parameters of fermentation kinetics were calculated: the lag time, the maximum rate of fermentation (Rmax), the time to reach Rmax, the $CO_2$ produced at Rmax, the rate of fermentation when 80 g/L $CO_2$ was produced and the time to achieve 80 g/L $CO_2$ production. The lag time of fermentation was chosen here as the time to achieve 1 g/L $CO_2$ production. All of these parameters were extracted from the weight loss data using the in-house R package alfisStatUtilR (v1.0.0) based on a local regression model.

## 2.6. Sample Preparation

All samples were collected at 80 g/L $CO_2$ production. For either metabolite or volatile compound analysis, a 5-mL sample was harvested using a cold 15-mL Falcon tube. The samples were centrifuged for 10 min at 3000 rpm at 4 °C, and the supernatants were stored at −20 °C until analysis. For the samples to be used for thiol analysis, 50 mg/L $SO_2$ was added to the sample to limit thiol oxidation during storage. The samples were centrifuged for 10 min at 3000 rpm at 4 °C, and the supernatant was stored at 4 °C for a maximum of 2 days before the analysis.

## 2.7. Determination of Metabolite Concentrations

Ethanol, glycerol, succinate and acetate concentrations were determined by HPLC (HPLC 1290 Infinity, Agilent Technologies, Santa Clara, CA, USA) in a Phenomenex Rezex ROA column (Agilent Technologies, Santa Clara, CA, USA) at 60 °C, as described by [12]. The column was eluted with 0.005 N $H_2SO_4$ at a flow rate of 0.6 mL/min. The acetic acid concentration was determined with a UV meter at 210 nm; the concentrations of the other compounds were determined with a refractive index detector. Analysis was carried out with the Agilent EZChrom software package.

## 2.8. Analysis of Fermentative Aromas by Gas Chromatography/Mass Spectrometry

The analysis of fermentative aromas was performed following the method of [12]. First, the volatiles were extracted with dichloromethane. Then, the concentrations of fermentative aromas were measured via GC/MS in SIM mode using a DB-WAX GC column. Thirty-five compounds were quantified using internal deuterated standards.

## 2.9. Thiol Analysis

All thiol analyses were performed by the Nyseos Society (Montpellier, France) using UPLC-MS/MS and stable isotope dilution assays. 3-mercaptohexan-1-ol (3MH) and 3-mercaptohexyl acetate (3MHA) analysis was performed following the method of [28]. 4-mercapto-4-methylpentan-2-one (4MMP) analysis was performed following the method of [29].

## 2.10. Statistical Analysis

All the experiments were carried out in biological duplicate.

Statistical analyses were performed with R version 3.2.3 (R Development Core Team 2016). Fermentation kinetics duplicates were smoothed using the in-house R package alfisStatUtilR (v1.0.0) based on the R locfit package (v1.5–9.1). To study aroma production in the different samples, heatmaps were plotted using the heatmap2 function from the R package gplots (v3.0.1.1).

## 3. Results

In the current work, we compared the effect of the addition of a "classic" assimilable nitrogen source (ammonium + amino acids) with that of peptide addition on the main fermentation kinetics and aroma synthesis.

To mimic peptide supplementation, we added peptides resulting from BSA digestion. Obviously, the use of these peptides is prohibited in oenology. More generally, the use of peptides is not legal

in oenology (as a result of E.U. legislation or O.I.V. rules). Consequently, the objective of our work is to constitute a proof of concept allowing to demonstrate the role of peptides during the alcoholic fermentation. The obtained results will not be directly applicable; it will be necessary to confirm and deepen the data generated in this study.

### 3.1. Effect of the Nitrogen Sources on Fermentation Kinetics

First, we compared the effect of nitrogen addition on the kinetic parameters of fermentation carried out with different yeast strains in Sauvignon Blanc must under the Control, YAN and Peptide conditions (Figure 2). It appears that all the yeast strains responded in the same way to the different nitrogen conditions, showing that the nitrogen effect was much greater than the strain effect regarding fermentation kinetics. Accordingly, we found that, for all the strains tested, the maximum fermentation rate (Rmax) was lowest for the Control condition, whereas the conditions involving nitrogen addition increased Rmax. These results are in accordance with numerous works showing that higher nitrogen concentrations lead to a higher Rmax [13,24,30,31]. More surprisingly, the highest value of Rmax was obtained under the Peptide condition for all the strains, despite the fact that the amount of assimilable nitrogen was the same in the YAN and Peptide conditions. As observed in the study of [10], we found that Rmax was reached later under the Peptide condition compared to the two other tested conditions. In line with the same previous study, we also found that the lag phase was slightly longer for the Peptide condition than for either the Control or YAN condition.

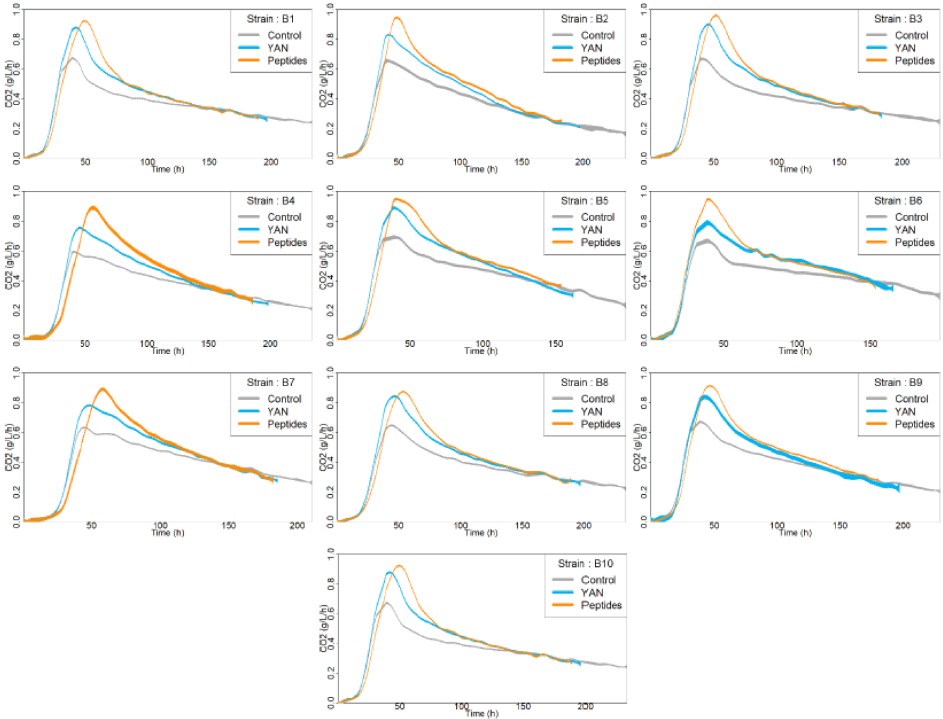

**Figure 2.** Fermentation kinetics of the Sauvignon for the 10 *S. cerevisiae* strains tested at 18 °C for three nitrogen conditions: the control condition, the YAN condition (ammonium + amino acids) and the Peptides condition (ammonium + amino acids + peptides).

All of these observations obtained during Sauvignon Blanc fermentation were also obtained during Merlot fermentation, which were performed using different *S. cerevisiae* strains (Figure 3).

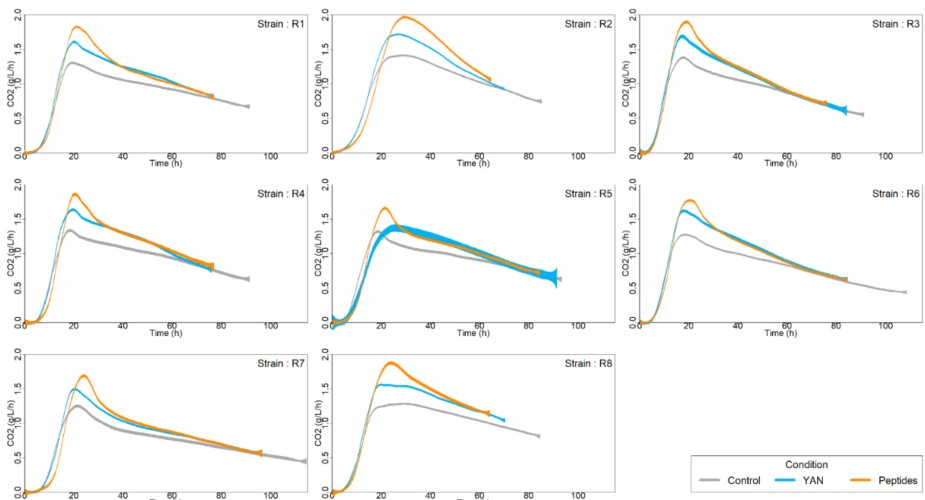

**Figure 3.** Fermentation kinetics of the Merlot for the 8 *S. cerevisiae* strains tested at 26 °C for three nitrogen conditions: the control condition, the YAN condition and the Peptides condition.

To obtain more accurate information on the different parameters of fermentation kinetics (i.e., time of the lag phase, Rmax, time at Rmax, $CO_2$ produced at Rmax, fermentation rate at 80 g/L $CO_2$ production, time to achieve 80 g/L $CO_2$ production), we used the R software to extract these parameters from the fermentation kinetics. Moreover, as the main objective was to compare the effect of the Peptide and YAN conditions, we calculated the ratios between Control and YAN conditions and between the Control and Peptide conditions. The obtained results were hierarchically classified using the R heatmap2 function and are shown in Figure 4.

For Sauvignon Blanc and Merlot fermentation (Figure 4A,B, respectively), we found two clusters that allowed the differentiation of the Peptide and YAN conditions according to the six estimated parameters. Relative to the Control, the values of all the kinetic parameters were higher under the Peptide than the YAN conditions, which logically resulted in a shorter time to reach 80 g/L $CO_2$ production under the first condition (reduction of 15 h on average). For Merlot fermentation, the fact that strains R5 and R7 grown under the Peptide condition were located in the cluster mainly associated with the YAN condition seemed to be due to the fermentation rate at 80 g/L $CO_2$ production, which was relatively low under the Peptide condition.

### 3.2. Effect of Nitrogen Sources on Yeast Fermentation Byproducts during Fermentation

We compared the effect of nitrogen addition on the production of metabolites issued from the central carbon metabolism by the strains used for white and red fermentation. We measured the concentrations of these byproducts at 80 g/L $CO_2$ production. For these molecules, we calculated the ratio between the two fermentation conditions with nitrogen addition (YAN and Peptide) and the Control condition (Figure 5). The analysis clearly distinguished the two nitrogen conditions under both Sauvignon Blanc (Figure 5A) and Merlot (Figure 5B) fermentation.

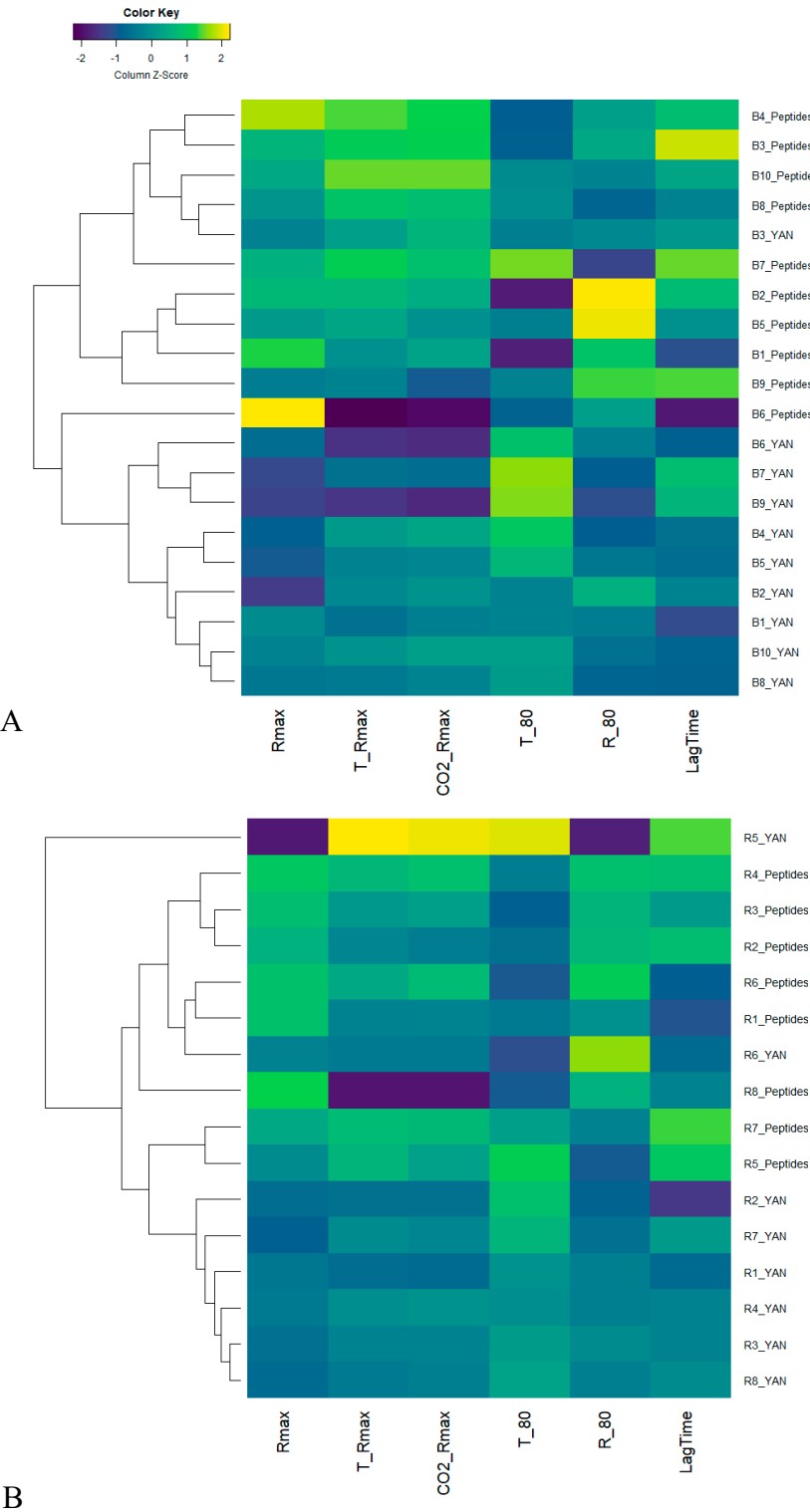

**Figure 4.** Heatmaps of the relative kinetic parameters for the fermentation of Sauvignon (**A**) and Merlot (**B**) for the *S. cerevisiae* strains tested. For each strain, YAN refers to the ratio between the YAN condition and the Control; Peptides refers to the ratio between the Peptides condition and the Control. Rmax: maximum rate fermentation; T_Rmax: time to reach the Rmax; CO2_Rmax: $CO_2$ produced when Rmax is reached; T_80: time to reach 80 g/L of $CO_2$ produced; R_80: rate of fermentation at 80 g/L of $CO_2$ produced; LagTime: time to reach 1 g/L of $CO_2$ produced.

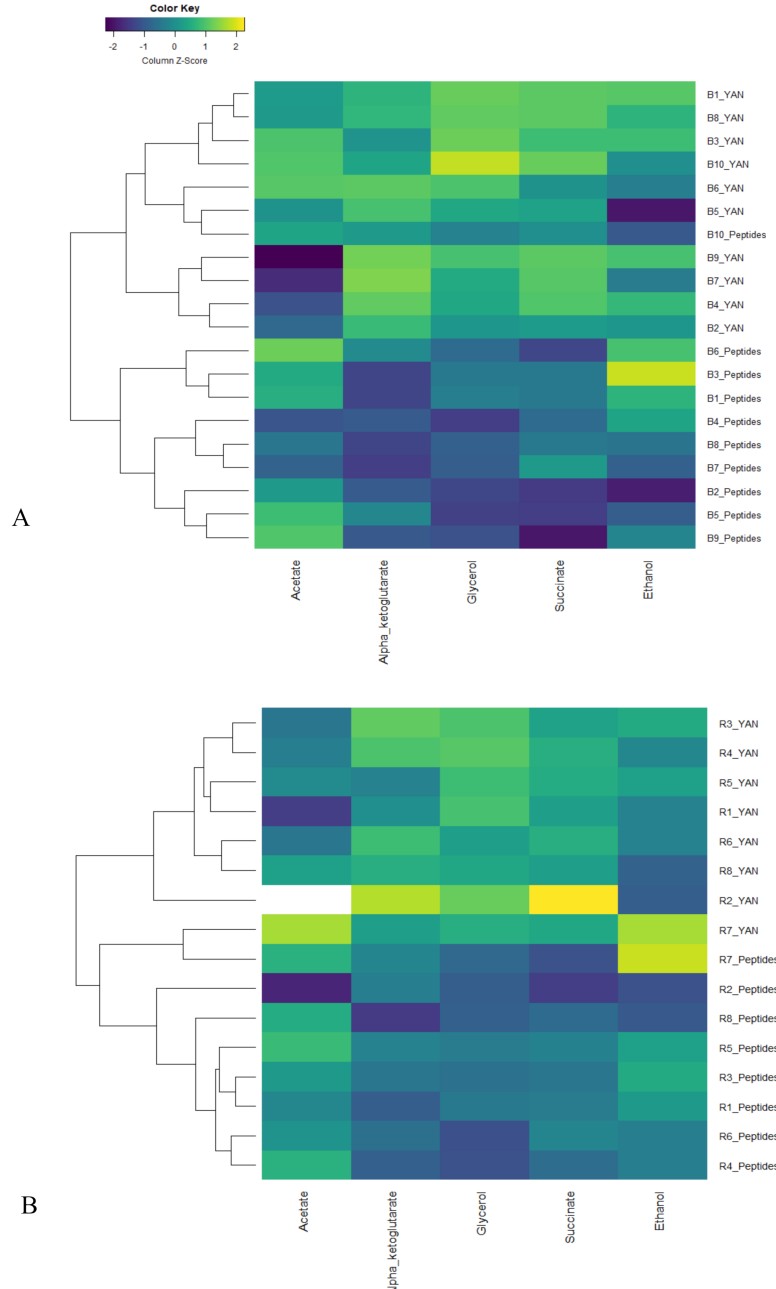

**Figure 5.** Heatmaps of the relative production of metabolic organic acid for the Sauvignon fermentation (**A**) and the Merlot fermentation (**B**) for the *S. cerevisiae* strains tested. For each strain, YAN refers to the ratio between the YAN condition and the Control; Peptides refers to the ratio between the Peptides condition and the Control.

Contrary to the study of [10], which showed a lower production of acetate when more peptides had been assimilated by yeast, we found that under our Peptide condition, yeasts produced a higher concentration of acetate compared to that under the YAN condition (54% increase on average). The same result was found under both Sauvignon Blanc and Merlot fermentation. For both types of must, succinate production appeared to be lower under the Peptide condition (12% decrease on average).

### 3.3. Effect of Nitrogen Sources on Fermentative Aroma and Thiol Production during Fermentation

We compared the effect of nitrogen addition on the production of aromas by the strains used for white and red fermentation. We measured the concentrations of the main fermentative aromas

that impact the organoleptic quality of the wine as well as the main thiols. We thus measured the concentrations of higher alcohols, ethyl esters, acetate esters, 4MMP, 3MH and 3MHA.

For Sauvignon Blanc and Merlot fermentation, when we considered all the compounds together (Figure 6), it appeared that the two nitrogen conditions were clearly separated (except for yeast strain B7 under the YAN condition in Sauvignon Blanc fermentation and strain R7 under the YAN condition in Merlot fermentation). This result shows that nitrogen sources have a specific impact on the production of aroma compounds. It therefore seems that the nature of the added nitrogen clearly differentiates the aroma profile of red and white wines.

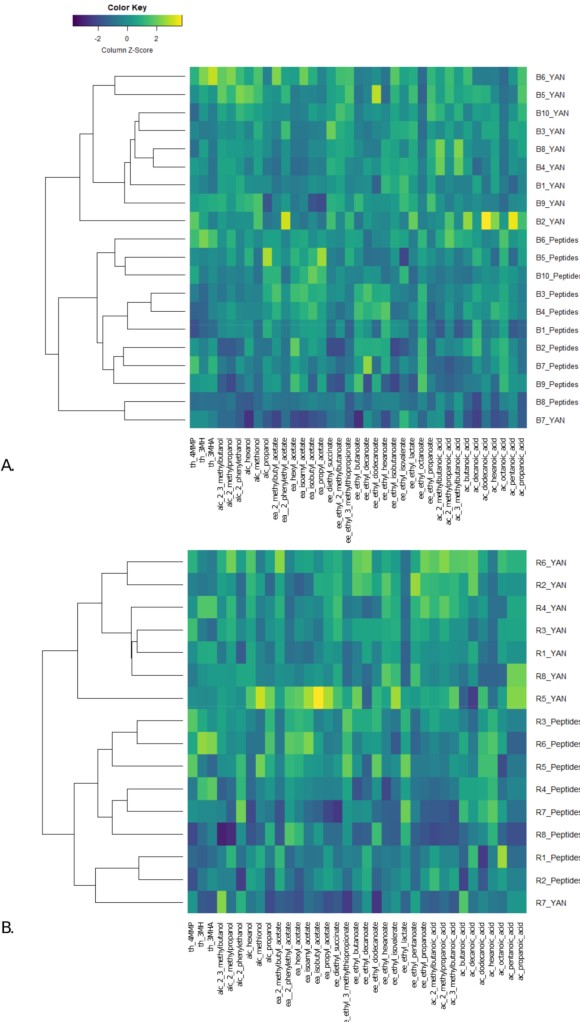

**Figure 6.** Heatmap of the relative production of fermentative aromas and thiols during the Sauvignon (**A**) and the Merlot (**B**) fermentations by the *S. cerevisiae* strains tested. For each strain, YAN refers to the ratio between the YAN condition and the Control; Peptides refers to the ratio between the Peptides condition and the Control.

In Sauvignon Blanc fermentation (Figure 7), the two nitrogen conditions were clearly separated (with the exception of one or two strains) as a function of the different groups of fermentative aromas (i.e., higher alcohols, ethyl esters and acetate esters). For the higher alcohols (Figure 7B), it appears that, compared to the Control condition, the addition of YAN led to higher production of these compounds than the addition of peptides (19% decrease on average). Only propanol seemed to be overproduced in the Peptide condition (17% increase on average). The ethyl esters (Figure 7C) were separated into two clusters: one for the ethyl esters that were overproduced under the YAN condition

(ethyl-2-methylbutanoate, ethyl isobutanoate, ethyl propanoate, diethyl succinate, ethyl lactate, ethyl isovalerate, ethyl-3-methylthiopropionate; overproduction of 15% on average) and another for those that were overproduced under the Peptide condition (ethyl butanoate, hexanoate, octanoate and decanoate; overproduction of 18% on average). For acetate esters (Figure 7D), overproduction appeared to occur under the Peptide condition (18% increase on average), with the exception of 2-phenylethyl-acetate). Regarding thiol production (Figure 7A), it was not possible to observe differences between the YAN and Peptide conditions, as the classification led to several clusters that formed under either the Peptide or YAN condition. For this last group of compounds, the strain effect appeared to be stronger than the effect of nutrition.

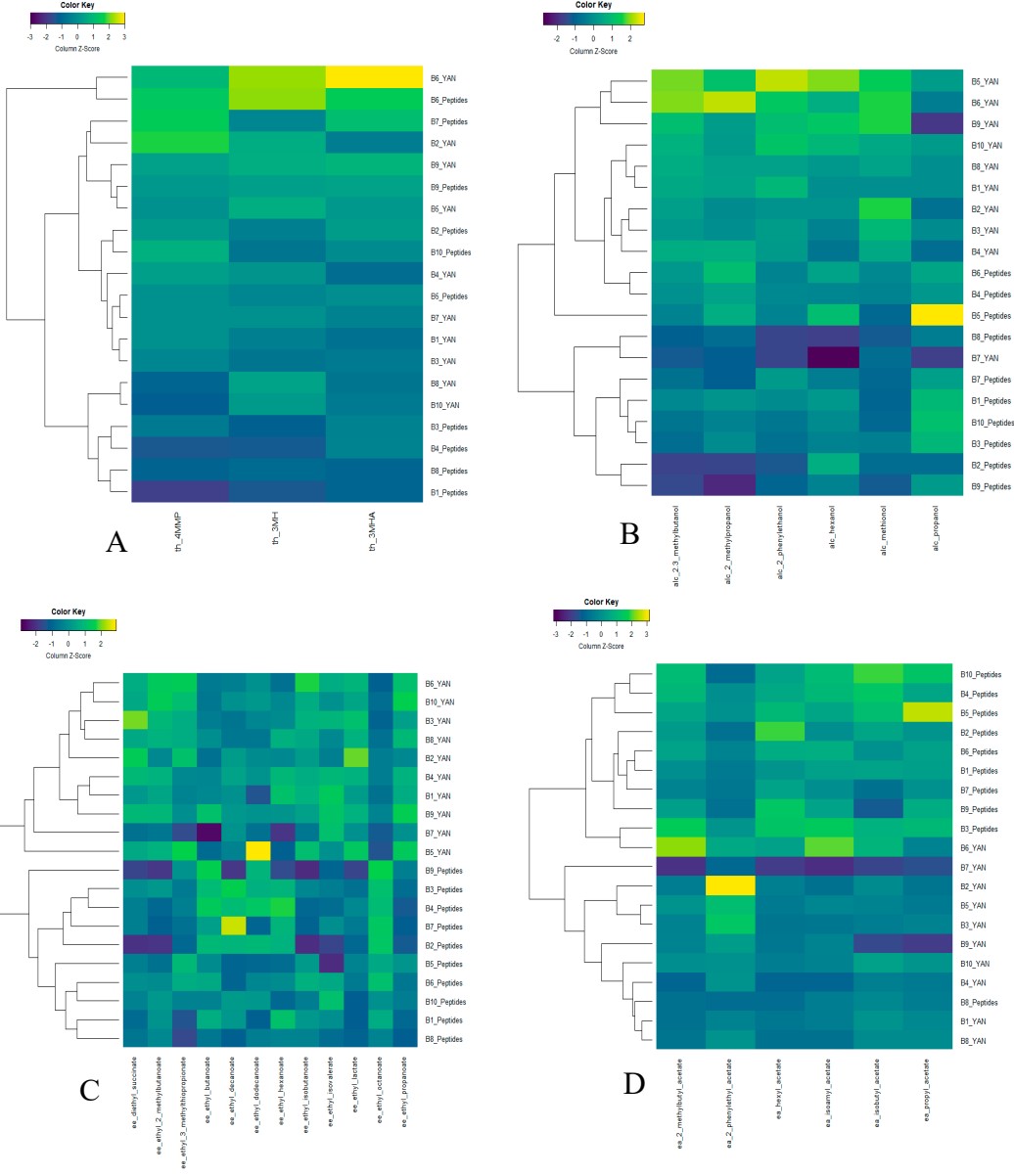

**Figure 7.** Heatmap of the relative production of thiols (**A**), higher alcohol(**B**), ethyl esters (**C**) and acetate esters (**D**) during the Sauvignon fermentations by the *S. cerevisiae* strains tested. For each strain, YAN refers to the ratio between the YAN condition and the Control; Peptides refers to the ratio between the Peptides condition and the Control.

For Merlot fermentation (Figure 8), the results were quite different. Indeed, it appears that a distinction between the YAN and Peptide conditions was clear only for ethyl esters (Figure 8C).

The clustering of the higher alcohols (Figure 8B) and acetate esters (Figure 8D) did not allow the separation of fermentation conditions according to the nature of the added nitrogen. For acetate esters (Figure 8D), it appeared that some strains (such as R1, R2, R7 and to a lesser extent R4) showed similar behavior regardless of the nature of the nitrogen added. As for Sauvignon Blanc fermentation, thiol production (Figure 8A) seemed to be more affected by the strain than by the nature of the nitrogen added.

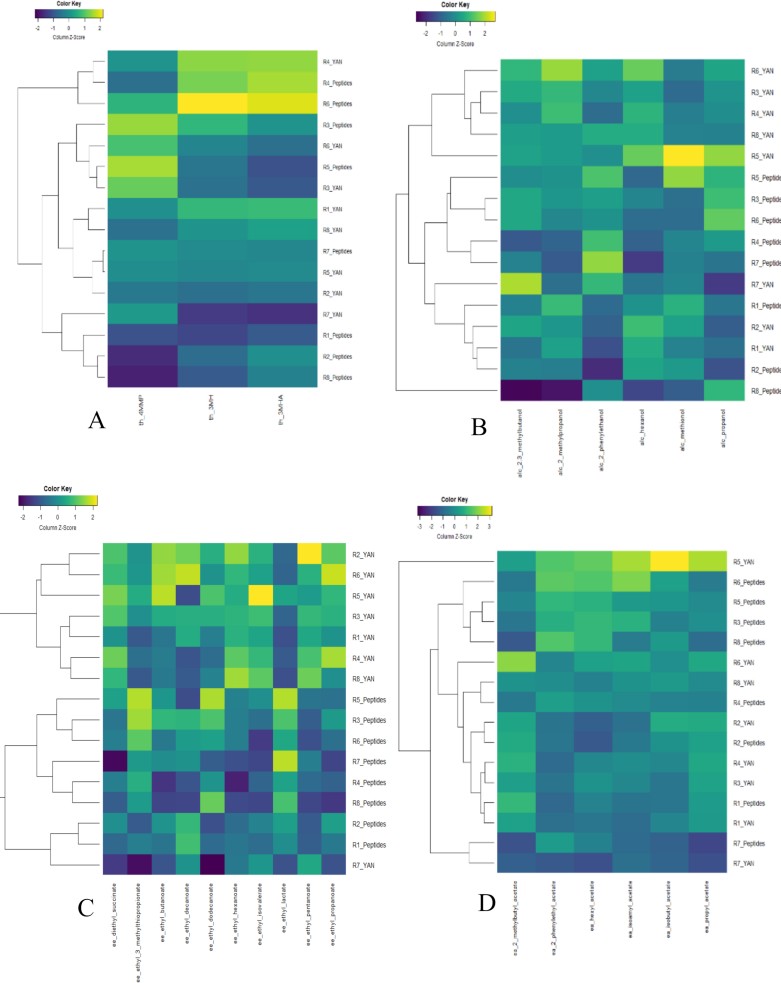

**Figure 8.** Heatmap of the relative production of thiols (**A**), higher alcohol(**B**), ethyl esters (**C**) and acetate esters (**D**) during the Merlot fermentations by the *S. cerevisiae* strains tested. For each strain, YAN refers to the ratio between the YAN condition and the Control; Peptides refers to the ratio between the Peptides condition and the Control.

## 4. Discussion

Peptides, particularly the small peptides contained in grape must, are almost never taken into account when measuring yeast-assimilable nitrogen. Although transporters of di-, tri-, tetra- and penta-peptides are well known [7,8,32,33] and it has been recently shown that the proteins encoded by the genes *FOT1* and *FOT2* are involved in the transport of oligopeptides of up to nine amino acids, it is still not clear how yeast use these peptides. The impact of these nonconventional nitrogen sources on enological fermentation is also poorly understood. This work aimed to differentiate the impact on the parameters of fermentation kinetics and the wine aroma profile between the addition of a classic nitrogen source (i.e., a mixture of amino acids and ammonium) and a more complex one containing peptides (of up to nine amino acids).

We focused our study on these two kinds of nitrogen addition and used an automatic approach to assess kinetic parameters and a heatmap representation to more easily observe the differences in aroma production.

A clear differentiation of parameters of fermentation kinetics was found between the addition of the classic nitrogen form (YAN) and the small peptides when the same amount of nitrogen was provided at the beginning of fermentation, regardless of the fermentation conditions (for white or red wine) and the strains used. It appeared that when peptides were added, Rmax was higher, and the fermentation time was shorter. Moreover, in the presence of peptides, Rmax was reached later, and the lag time of fermentation was longer. This last observation is in accordance with those of [10]: in this specific study, a longer lag time and a later Rmax were obtained when using a strain exhibiting the Fot1/2 peptide transporters in addition to the endogenous, Ptr2, Dal5, Opt1 and Opt2 transporters (i.e., showing the ability to use a broader range of oligopeptides) compared to a strain exhibiting only the endogenous transporters. Interestingly, these results were found during both Sauvignon Blanc and Merlot fermentation involving the 18 *S. cerevisiae* strains tested. This suggests that the impact of peptides on the parameters of fermentation kinetics is not strain dependent. All of these results showed that the nature of the nitrogen added leads to specific effects on the parameters of fermentation kinetics and that the addition of a nitrogen source with a significant proportion of peptides seems to increase fermentation kinetics. Interestingly, the same results have been observed in the context of beer fermentation under soy peptide addition [34], showing that the effect of peptide addition appears to be more general, rather than being restricted to oenological conditions.

These specific effects of the addition of peptides compared to classic nitrogen sources could also be observed in the production of molecules issued from the central carbon metabolism. Indeed, we found that these compounds were produced in lower quantities under peptide addition compared to classic nitrogen addition in both and or red fermentation. This could reflect a modulation of the central carbon metabolism by the peptides once more small peptides have been assimilated, as proposed by [10]. Additionally, as found by [12], nitrogen availability modulates the production of these byproducts. Moreover, the assimilation of a broader range of peptides via Fot1/2 transporters modulates nitrogen metabolism [10]. Thus, the differences that we observed might reflect a modulation of nitrogen management by the cells under peptide addition, as it has been observed between ammonium addition and amino acid addition [13].

For acetic acid, we observed overproduction under peptide addition. This variation in acetate production together with that of the other fermentation byproducts seems to indicate a reorganization of the redox metabolism of the yeast, as [10] likely arising from changes in NADPH pool management. Similarly, the lower production of succinic acid and alpha-ketoglutarate observed under peptide addition compared to YAN condition seems to show reorganization in the management of the glutamate node [12]. Finally, an important point is that although changes in the production of acetate were observed both in our study and in the work of [10], the direction of variation was different between the two studies. This difference probably arises from the fact that the nature of the added peptides and, therefore, their amino acid composition are different in the two research works.

Considering the production of aromas, the effects of the nitrogen form were different between the fermentation of white and red musts. Indeed, in Sauvignon Blanc fermentation, the impact of peptide addition on the three main fermentative aroma families (higher alcohols, acetate esters and ethyl esters) was clear, whereas no such effect was observed in Merlot fermentation.

Thus, regarding Merlot fermentation, no clear distinction between classic nitrogen addition and peptide addition could be identified in terms of acetate esters or higher alcohols. This low differentiation of the effect of the nitrogen form on aroma production could be explained by the higher complexity of the red wine must (higher polyphenol content, for example). Another explanation for the variability in the effect of the nitrogen form on aroma production could be partly due to the genetic background of the strains used in our study, which were different from those used for white wine fermentation.

However, an important point is that ethyl esters were the only aroma family for which a distinction between peptide addition and classic nitrogen could be made in both Merlot and Sauvignon Blanc fermentation. These results therefore reflect a major effect of peptide addition on the synthesis of these aromas, even though this effect was somewhat unexpected. Indeed, several reports have shown only a few changes in the production of ethyl esters by yeast upon the addition of nitrogen during fermentation [12,13] when "classic" nitrogen sources (ammonium or amino acids) are added. Ethyl ester production is indeed closely related to fatty acid synthesis, as these aromas result from the addition of ethanol to fatty acids by an acyltransferase [35]. In our research work, the impact of peptides on the synthesis of ethyl esters appeared to be an indirect effect linked to changes in redox metabolism. Indeed, the production of acetyl-CoA and fatty acids, which are precursors of ethyl esters, could be impacted by the management of the NADPH pool [36,37]. This appears to be consistent with our previous observations on fermentation byproducts suggesting a reorganization of the central carbon metabolism. Indeed, under peptide addition, we observed the overproduction of acetate, which is one of the precursors of acetyl-CoA through pyruvate dehydrogenase (PDH) bypass [38]. This variation in the management of acetyl-CoA could thus be related to the observed difference in the production of ethyl esters [10].

All of these modifications of pools of intracellular metabolites and redox balances could also be consistent with the results regarding the synthesis of another aroma family observed in white wine fermentation. Indeed, we found higher production of acetate esters under peptide addition. As this aroma family requires acetyl-CoA for its production [15], the variation in its production may result from the reorganization of the management of the NADPH and acetyl-CoA pools discussed previously. Moreover, acetate esters are known to be impacted by nitrogen availability [12,13]. Their global overproduction under peptide addition suggests a change in the regulation of the alcohol transferases *ATF1/2*, which are key genes for the synthesis of acetate esters [39]. Indeed, in the work of [13], it has been shown that overexpression of *ATF1/2* occurs under the late addition of nitrogen, especially in the form of amino acids. Moreover, it has been shown that yeast preferentially use oligopeptides, leading to later use of free amino acids in the must [10]. It is thus possible that peptide addition in the present study led to the later use of free amino acids. This later presence of free amino acids could thus induce *ATF1/2* overexpression, explaining the higher concentration of acetate esters observed.

All of the examined higher alcohols were produced in lower quantities under peptide addition with the exception of propanol. This lower production of higher alcohols may result from changes in the management of the pool of keto-acids, which are precursors of these compounds. This last observation is in line with the lower production of succinic acid observed, which is a key compound in the TCA cycle and can be considered a safety valve to prevent the accumulation of $\alpha$-ketoglutarate [12]. Interestingly, 2-phenylethanol was underproduced under peptide addition. This specific alcohol is produced through the pentose phosphate pathway, which leads to high production of NADPH. This finding could thus be consistent with the reorganization of the management of the NADPH pool discussed previously. Finally, the overproduction of propanol in the presence of peptides seems to reflect the fact that propanol can be considered a marker of nitrogen quality [13] and not only of nitrogen availability [12,40]. This last result confirms the particular role of propanol compared to the other higher alcohols and its atypical management by yeast.

Regarding the thiols that are not directly produced via yeast metabolism but are released from thiol precursors, no differentiation was found between the two kinds of nitrogen addition. This revealed that the main factor affecting the production of these volatile compounds is the strain effect. This observation is consistent with numerous studies showing the importance of the yeast genetic background for thiol release [19,41].

## 5. Conclusions

In conclusion, through the use of an innovative online monitoring system and associated automatic data treatment "pipeline", we found that peptide addition at the beginning of wine alcoholic

fermentation has an impact on the parameters of fermentation kinetics as well as the aroma profile. First, it was shown that the addition of oligopeptides induces a shortening of the fermentation duration, mainly due to a higher maximum fermentation rate. Second, differences in the synthesis of compounds linked to the central carbon metabolism (mainly acetate) or aromas (mainly ethyl esters) suggest that peptide addition induces significant changes in the management of the intracellular pool of NADPH. Finally, considering volatile compounds, the effect of peptides was different depending on the studied compounds: it was very low for thiols and much more important for fermentation aromas. Moreover, impact of peptide addition was obvious under Sauvignon Blanc fermentation compared to Merlot fermentation, except for ethyl esters, which were overproduced in all tested conditions. It appears that the addition of peptides leads to a change in the concentrations of higher alcohols, acetate esters and ethyl esters involved in wine aroma. Following a peptide addition, quantities and balances between the volatile compounds belonging to these three chemical families were very different from the ones observed under the addition of classic yeast-assimilable nitrogen sources (i.e., amino acids and ammonium). Nevertheless, our research only constitutes a proof of concept of the importance of peptides in alcoholic fermentation. Deeper studies will be necessary to detail the impact of these molecules on the aroma genesis. In particular, the ability to measure the peptide contents of the must would be of great help in adjusting the nitrogen supply during fermentation according to the type of aromas desired.

**Author Contributions:** Conceptualization, C.D., V.M., S.D., A.S. and J.-R.M.; formal analysis, C.D. and I.S.; investigation, C.D., F.M. and S.D.; methodology, C.D., F.M. and V.G.; writing—original draft preparation, C.D., I.S., V.G., S.D., A.S. and J.-R.M.; writing—review and editing, C.D., V.G. and J.-R.M. All authors have read and agreed to the published version of the manuscript.

**Funding:** This work was funded by the Single Inter-Ministry Fund (FUI) of the NV$^2$ project.

**Acknowledgments:** We thank Valérie Nolleau for her assistance with the aroma analysis, as well as Marc Perez and Christian Picou for their support in using the automated fermentation platform. We thank Carole Camarasa for the helpful discussion on peptide and amino acid metabolism in yeast. The authors would like to thank the partners of the NV$^2$ project, notably itk, Frayssinet, Lallemand and Nyseos.

**Conflicts of Interest:** The authors declare no conflict of interest. The funders had no role in the design of the study; in the collection, analyses, or interpretation of data; in the writing of the manuscript, or in the decision to publish the results.

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
