# Peer review of "Large-Scale Screening of Thiol and Fermentative Aroma Production during Wine Alcoholic Fermentation: Exploring the Effects of Assimilable Nitrogen and Peptides"

_fermentation, doi:10.3390/fermentation6040098_

Round 1

Reviewer 1 Report

Review comments on large-scale screening of thiol and fermentative aroma production during winemaking fermentation: effect of assimilable nitrogen and peptides

The study focused on the effect of peptides on the fermentative kinetics and production of aroma to better understand the impact of nitrogen compounds on the metabolism of S. cerevisiae. Extensive research was conducted as seen from the large number of parameters analyzed. The authors may have omitted one of the graphs for the 10th S. cerevisiae strain tested for the fermentation kinetics of the Sauvignon (Fig. 2).

The use of heat map in data representation offered the authors the benefit of evaluating many parameters and at the same time, provided on the spot view to easily compare the factors and the parameters being assessed.

However, the heat map mostly offered qualitative assessment for the various parameters compared between YAN and peptides instead of quantitative assessment. For instance, the difference in the data for the factors being compared (i.e. YAN and peptides) were described as being higher or lower, longer or shorter and so on. It would be more instructive if the authors would provide more quantitative information such as how much did the peptide surpass the YAN to reach 80 g/L CO2 production (if possible). The same goes for other parameters compared, if convenient.

The authors concluded that the addition of peptides during wine fermentation using S. cerevisiae enhanced the fermentation rate and the synthesis of compounds linked to central carbon metabolism such as acetate. Although, from the heat map in Fig. 5, the five parameters being evaluated generally had higher output for YAN than for peptides. The same goes for other figures such as Fig. 6 and 7 showing fermentative aromas and thiol production. More so, it is better to clearly state the parameters in which the addition of peptides outperformed the YAN and the relevance of those enhanced parameters in wine fermentation and processing in general.

Author Response

The study focused on the effect of peptides on the fermentative kinetics and production of aroma to better understand the impact of nitrogen compounds on the metabolism of S. cerevisiae. Extensive research was conducted as seen from the large number of parameters analyzed. The authors may have omitted one of the graphs for the 10th S. cerevisiae strain tested for the fermentation kinetics of the Sauvignon (Fig. 2).

In Fig. 2, the graph corresponding to the 10th S. cerevisiae strain was added.

The use of heat map in data representation offered the authors the benefit of evaluating many parameters and at the same time, provided on the spot view to easily compare the factors and the parameters being assessed.

However, the heat map mostly offered qualitative assessment for the various parameters compared between YAN and peptides instead of quantitative assessment. For instance, the difference in the data for the factors being compared (i.e. YAN and peptides) were described as being higher or lower, longer or shorter and so on. It would be more instructive if the authors would provide more quantitative information such as how much did the peptide surpass the YAN to reach 80 g/L CO2 production (if possible). The same goes for other parameters compared, if convenient.

The percentages of decrease or increase have been specified in brackets in the result section.

The authors concluded that the addition of peptides during wine fermentation using S. cerevisiae enhanced the fermentation rate and the synthesis of compounds linked to central carbon metabolism such as acetate. Although, from the heat map in Fig. 5, the five parameters being evaluated generally had higher output for YAN than for peptides.

The addition of peptides enhanced the fermentation rate and reduced the fermentation time, as shown on Fig. 2 and 3. Concerning, the central carbon, compared to YAN, the addition of peptides provoked a 54% increase for acetate on average whereas a 12% decrease on average was observed for both succinate and glycerol. This information was added at the end of section 3.2.

The same goes for other figures such as Fig. 6 and 7 showing fermentative aromas and thiol production. More so, it is better to clearly state the parameters in which the addition of peptides outperformed the YAN and the relevance of those enhanced parameters in wine fermentation and processing in general.

This observation was added in section 3.3.

Reviewer 2 Report

The manuscript “Large-scale screening of thiol and fermentative aroma production during winemaking fermentation: effect of assimilable nitrogen and peptides” by Camille Duc and colleagues show a research work about the use of small peptides addition during the wine alcoholic fermentation and their impact on fermentation kinetics and aroma production by the use of different yeast strains for Sauvignon blanc and Merlot fermentations. The results, were compare by the use of classic assimilable nitrogen sources. It is important to note that the small peptides addition during wine fermentation is not a legal practice and this point must be highlighted, especially at the end of introduction. However, the topic of the manuscript its interesting and give us relevant information. In addition, this manuscript is well written and the experimental procedure is generally adequate.

In my opinion, there are only a few points that should be change/revise before a possible publication.

Comments:

Title - This work is a previous work. Thus, in my opinion, it will be necessary to introduce this mention.

Abstract - Last sentence must be move for the end of Conclusions or Introduction items.

Line 72 - Change “volatile compound production ....” for ““volatile compound production during alcoholic fermentation.”

Line 119-128 (figure 1) - This is an explanation of the figure. Thus, this point must be introducing directly in the text (for example in item 2.4.) and not as a legend.

Line 144-149 - Reference about the HPLC methodology must to be introduce (where was described for the first time or validated).

Line 150-154 - More details are necessary for the GC-MS conditions and methodology.

 Line 168-170 - The legal aspects of the peptide’s addition it’s e relevant point. Thus, this point must be reintroducing in the introduction clearly and objectively. In addition, the use of peptides is prohibited in oenology as a result of E.U. legislation or O.I.V. rules ? The use of this practice is not legal in all wine production countries ? This point must be well clarified.

Figure 2 - Introduce in parentheses the main conditions for “YAN condition” and “Peptides condition”. It will be clearer to the readers.

Discussion and conclusions:

Could be interesting to introduce some discussion about the potential role of alcoholic fermentation temperature on peptide effects.

Line 287-298 - I suggest to move these sentences for the introduction, where is showing the relevance and novelty of this study.

Line 326-329 - Rewrite the sentence. The position of the reference it is not clear.

Line 336-339; 396-397 - This is true. However, it is not showing the individual volatile compounds concentration obtained and consequently we don’t know if the values correspond to concentrations above or below the detection thresholds of the compounds. Thus, it will not be correct to mention that the aroma profile obtained was totally different.

Line 340-346 - Probably, some phenolic compounds has an antimicrobiological activity for several yeast strains and consequently the yeast metabolism could be affected.

Line 388-392 - There seems to be some contradiction between the sentences. I suggest to rewrite.

Author Response

The manuscript “Large-scale screening of thiol and fermentative aroma production during winemaking fermentation: effect of assimilable nitrogen and peptides” by Camille Duc and colleagues show a research work about the use of small peptides addition during the wine alcoholic fermentation and their impact on fermentation kinetics and aroma production by the use of different yeast strains for Sauvignon blanc and Merlot fermentations. The results, were compare by the use of classic assimilable nitrogen sources. It is important to note that the small peptides addition during wine fermentation is not a legal practice and this point must be highlighted, especially at the end of introduction. However, the topic of the manuscript its interesting and give us relevant information. In addition, this manuscript is well written and the experimental procedure is generally adequate.

As requested by the reviewer, it was precised at the end of the introduction that adding peptides alone is not a legal practice.

In my opinion, there are only a few points that should be change/revise before a possible publication.

Comments:

Title - This work is a previous work. Thus, in my opinion, it will be necessary to introduce this mention.

As requested by the reviewer, the title was modified.

Abstract - Last sentence must be move for the end of Conclusions or Introduction items.

The last sentence of the abstract was moved to the end of the introduction.

Line 72 - Change “volatile compound production ....” for ““volatile compound production during alcoholic fermentation.”

This change was done.

Line 119-128 (figure 1) - This is an explanation of the figure. Thus, this point must be introducing directly in the text (for example in item 2.4.) and not as a legend.

This part was suppressed and moved to the paragraph 2.3.

Line 144-149 - Reference about the HPLC methodology must to be introduce (where was described for the first time or validated).

A reference was added.

Line 150-154 - More details are necessary for the GC-MS conditions and methodology.

Details for the GC-MS conditions and methodology are presented in the publication [12] of Rollero et al., 2015. So, I propose not to add information in the current paper. If the reader want to have the details, he can find them in this specific publication.

 Line 168-170 - The legal aspects of the peptide’s addition it’s e relevant point. Thus, this point must be reintroducing in the introduction clearly and objectively. In addition, the use of peptides is prohibited in oenology as a result of E.U. legislation or O.I.V. rules ? The use of this practice is not legal in all wine production countries ? This point must be well clarified.

This point was clarified.

Figure 2 - Introduce in parentheses the main conditions for “YAN condition” and “Peptides condition”. It will be clearer to the readers.

Compositions of “YAN” and “Peptides” conditions were added in Figures 2 and 3.

Discussion and conclusions:

Could be interesting to introduce some discussion about the potential role of alcoholic fermentation temperature on peptide effects.

We agree to the reviewer but, unfortunately, at present, discussing the impact of temperature on peptide metabolism is not possible. There is very little information regarding the links between peptides and aromas and no study has been carried out on the effect of temperature, to our knowledge.

Line 287-298 - I suggest to move these sentences for the introduction, where is showing the relevance and novelty of this study.

Unfortunately, it is not possible to find these sentences so it is impossible to do the action desired by the reviewer. There is probably a mistake in the number of the lines cited by the reviewer.

Line 326-329 - Rewrite the sentence. The position of the reference it is not clear.

This sentence was rewritten.

Line 336-339; 396-397 - This is true. However, it is not showing the individual volatile compounds concentration obtained and consequently we don’t know if the values correspond to concentrations above or below the detection thresholds of the compounds. Thus, it will not be correct to mention that the aroma profile obtained was totally different.

The corresponding sentence were changed: the term “aroma profile” was removed.

Line 340-346 - Probably, some phenolic compounds has an antimicrobiological activity for several yeast strains and consequently the yeast metabolism could be affected.

Line 388-392 - There seems to be some contradiction between the sentences. I suggest to rewrite.

These sentences were rewritten.

Round 2

Reviewer 1 Report

Authors have taken the time to improve on the manuscript. The edits reflect reviewers' comments. While the manuscript is greatly improved, a few minor edits are required before the manuscript can be published. 

For instance, the tile still needs to be tweaked. The use of "winemaking fermentation" is not appropriate. Authors may want to say "wine fermentation" or just stick to "wine making". Also, still on the title, it would be better to say "exploring the effects of..." rather than "exploration of the effects of..."

Author Response

Following the reviewer’s recommendations, the title was changed. The new title is: "Large-scale screening of thiol and fermentative aroma production during wine alcoholic fermentation: exploring the effects of assimilable nitrogen and peptides"